# Measuring and Leveraging Motives and Values in Dietary Interventions

**DOI:** 10.3390/nu13051452

**Published:** 2021-04-25

**Authors:** Sarah J. Eustis, Gabrielle Turner-McGrievy, Swann A. Adams, James R. Hébert

**Affiliations:** 1Cancer Prevention and Control Program, University of South Carolina, Columbia, SC 29208, USA; MCGRIEVY@mailbox.sc.edu (G.T.-M.); ADAMSS@mailbox.sc.edu (S.A.A.); JHEBERT@mailbox.sc.edu (J.R.H.); 2Department of Health Promotion, Education and Behavior, Arnold School of Public Health, University of South Carolina, Columbia, SC 29208, USA; 3College of Nursing, University of South Carolina, Columbia, SC 29208, USA; 4Department of Epidemiology and Biostatistics, Arnold School of Public Health, University of South Carolina, Columbia, SC 29208, USA

**Keywords:** dietary interventions, weight loss, health outcomes, adherence, non-adherence, food motivation, food choice

## Abstract

Why measure and leverage food motives and values? Every failure and every success in dietary change can be connected to motivation. Therefore, this research question naturally arises: How can food motives and values be measured and leveraged to improve diet outcomes from the individual to populations? There are four ways that food motives and values (FMVs) can assist researchers and health professionals. First, FMVs can help to create a personalized approach to dietary change. Second, FMVs can inform content for dietary interventions. Third, these FMV measures can be used in data analysis to elucidate differences in adherence and outcomes among participants. Fourth, public health nutrition messages can be tailored using information on FMVs. Each of these uses has the potential to further the literature and inform future efforts to improve diet. A central aim of our study is to provide specific examples and recommendations on how to measure and leverage FMVs. To do so, we reviewed 12 measures included in the literature citing the Food Choice Questionnaire by Steptoe, Pollard, and Wardle, which was identified as the earliest, highly cited article appearing under the search terms “food motives” AND “food values” AND “eating behavior” AND “measure”. Specific details on how articles were selected from the citing literature are described in the Methods section. We also expound on our reasoning for including the Three-Factor Eating Questionnaire, which made for 13 measures in total. Our main finding is that each measure has strengths and shortcomings to consider in using FMVs to inform nutritional recommendations at different levels.

## 1. Introduction: Challenges and Opportunities in Dietary Change

Diet has the potential to prevent, treat, and even reverse chronic health conditions [1,2,3,4,5,6]. The fact that diet underlies myriad health outcomes highlights the importance of promoting nutritious eating. Syndemic theory is useful for understanding the importance of diet, because the theory shifts focus from specific endpoints to disease interaction [7]. As diet affects risk for many types of disease, including infections, chronic illness, and mental health, promoting nutrition can have far-reaching effects for a wide variety of health outcomes [7,8,9]. Understanding why people eat what they eat can inform efforts to improve diet from the individual to population levels.

While dietary counselling or interventions range in duration from weeks to months, it make take years to establish and then maintain dietary changes [10,11,12,13,14]. Despite their potential to improve health, efforts aimed at dietary change are limited in important respects [15,16,17,18,19]. Adherence is a significant barrier for health professionals working with individuals or groups. For dietary researchers specifically, adherence poses a challenge because incomplete compliance results in overestimates of the actual dose of the dietary prescription while attenuating the measure of association (e.g., dose–response curve, relative risk estimate) [20,21]. This means that the effect of diet may be underestimated.

Why measure food motives to address long-term dietary change and low adherence? How? We can measure FMVs by examining explicit reasons for food choice or implicit influences on eating behavior (using the tools in the following section). Studying these factors is likely to be useful in understanding why some individuals fail to improve their diet, while others succeed (in research and real-world settings).

In the following sections, we describe the strengths, shortcomings, and potential utilities of a variety of tools for measuring motivations and values related to food and eating behavior. Then, we discuss past and potential future uses for individual counselling, intervention planning, study design, and widespread implementation and dissemination.

## 2. Methods: The Use of Food and Eating Behavior Questionnaires to Improve Dietary Interventions

As this article was intended to be a perspective on the appropriate use of measures of FMVs, and not to provide a systematic review of the literature meant to identify and catalog all articles addressing any aspect of food motivation or values, we approached this as follows. We first searched Clarivate Web of Science^®^ for the terms “food motives” AND “food values” AND “eating behavior” AND “measure”. From this, we found the earliest measure of FMVs was Steptoe, Pollard, and Wardle’s Food Choice Questionnaire (FCQ, published in 1995). This highly cited article (914 of 979 total citations as of 9 April 2021 according to Web of Science) enabled us to use the FCQ article as a “seed” article [22]. We then searched the titles of the citing literature (i.e., articles referencing the FCQ) for the following terms: “measure”, “tool”, “questionnaire”, “survey”, OR “scale”, which yielded 818 articles. Then, after excluding reviews, proceeding papers, and early access articles, 776 articles remained. From these articles, we excluded any that were repeats, that did not introduce a new measure, that did not pertain to food/eating behavior, that did not focus on motivations/values, or that were developed for use in a specific group or for specific foods. From the resulting 80 articles that potentially furthered the study aim to describe measures of FMVs and the applications, we identified 12 measures for inclusion. The Three-Factor Eating Questionnaire (TFEQ) does not cite the FCQ, but because the measure occupies an important place in the literature (it had been cited 3019 times as of 9 April 2021), it also was included [23]. In total, we review 13 articles. For a summary of the measures’ strengths, shortcomings, and applications, see Table 1.

### 2.1. The Dutch Eating Behavior Questionnaire

The Dutch Eating Behavior Questionnaire (DEBQ) contains scales for restrained eating (intense dieting and persistent hunger followed by excessive food intake), emotional eating (eating in response to internal cues), and external eating (eating in response to external cues) [24]. The measure was specifically developed for use in overweight and obese populations [24]. As the DEBQ was intended to elucidate eating patterns associated with obesity, it may be helpful for identifying eating patterns associated with poorer outcomes. Health professionals could also use the DEBQ to study associations among eating patterns and greater levels of non-adherence and attrition.

### 2.2. The Eating Motivation Survey (TEMS)

The TEMS is a comprehensive measure of motives relating to eating [25]. Renner et al., who developed the tool, intended to measure the motives that drive normal eating behavior, rather than focus on maladaptive health patterns in unhealthy samples (as with the DEBQ) [25]. The TEMS encompasses 15 factors represented by 78 items, which allows for a sensitive analysis of motives associated with food choice. Results may elucidate why individuals select certain foods or food types over others. As the tool measures a wide variety of motivators, it may be useful for developing more tailored approaches to dietary change and health messaging.

### 2.3. The Eating Motivations Scale

Recently developed by Raquel et al., the Eating Motivations Scale (EATMOT) provides a variety of determinants for people’s food choices, including health, emotions, price and availability, society and culture, environment and politics, and marketing and advertising [26]. The study was carried out in 16 countries from 2017 to 2018 and involved nearly 12,000 adult volunteers from various sociodemographic backgrounds. The final scale includes 20 questions that assess six categories of eating motives, making the measure short but useful. The scale contains items on health aspects, emotional status, economic and availability motivations, social and cultural influences, environmental and political determinants, and marketing and advertising [26]. The EATMOT scale provides information on general food motives that are not specific to certain groups or FMVs. The measure can be utilized for a variety of purposes, from individualized advice to public health efforts.

### 2.4. The Food and Beverage Need for Uniqueness Scale

Developed as a consumer tool, the Food and Beverage Need for Uniqueness Scale measures individuals’ propensity for unique food items [27]. The results may be helpful for measuring intolerance towards routine and familiarity in one’s diet. Findings denoting a high need for uniqueness can inform interactions with individuals and groups that prioritize food novelty. Health professionals (dieticians, nutrition interventionists, etc.) may prioritize allocation of new recipes, recommendations for uncommon food ingredients, and advice for maintaining variety. These actions could improve adherence to a nutritious eating pattern and enthusiasm for dietary change in some.

### 2.5. Food Choice Motives Questionnaire

The Food Choice Motives Questionnaire has more of a focus on sustainable food purchasing motives, including ethics, environment, and local production than comparable measures (see Section 2.6 The Food Choice Questionnaire and Section 2.8 Measure of Food Choice Values). The questionnaire contains 63 questions that cover nine dimensions: (1) ethics and environment, (2) local and traditional production, (3) taste, (4) price, (5) environmental limitations (i.e., not buying food out of environmental concerns), (6) health, (7) convenience, (8) innovation, and (9) absence of contaminants [28]. As with similar measures, the assessment provides information on the relative importance of each factor in food choice. This questionnaire expands on Lindeman and Vaananen’s ethics-focused addendum to the FCQ (See Section 2.9 Measurement of Ethical Food Motives). The Food Choice Motives Questionnaire covers concerns about the physical and social environment broadly, with items on production waste, packaging and pollution issues, fair trade productions, and respect for human rights and working conditions [28]. The authors highlight the importance of including these items, since interest in sustainability is growing [28]. The Food Choice Motives Questionnaire includes well-established food motives and emerging ones. The focus on purchasing behavior, rather than eating behavior, may be a shortcoming, because people may not eat the foods they buy. However, the survey remains a useful tool for research on consumer behaviors, which reflect food motives.

### 2.6. The Food Choice Questionnaire (FCQ)

The FCQ was developed in 1995 by Steptoe, Pollard, and Wardle [22], who recognized the initial gap in the literature on food and eating behavior. Until then, no tool had been developed to rigorously measure individuals’ motives surrounding food choice. They identified motives driving food choice through confirmatory factor analysis. Findings indicated that cost, convenience, familiarity, natural content (e.g., contains no additives, artificial ingredients), health, mood, sensory appeal, ethical concerns, and weight control were important motivating factors [22]. They found that these factors differ across individuals in terms of how they affect food choice [22]. These motivations may shift over time or vary by food encounter; nevertheless, primary motives can be identified with the FCQ, and these motives can provide useful information for health professionals and researchers.

Despite its strengths, the FCQ has limitations, including the lack of cross-cultural validation studies and the potential for social desirability and social approval to bias responses [35,36,37,38,39]. Though validated, scholars have questioned the comprehensiveness of the FCQ [31]. In other words, the FCQ has been criticized for failing to effectively capture all the possible motives for food choice. In a meta-analysis on the use of the FCQ internationally, Cuhna et al. found that the most common changes to the questionnaire made by researchers were additions [40]. On the other hand, researchers also have criticized the tool’s level of abstraction [40]. They suggested that many of the categorizations could be sensibly grouped (e.g., weight control and health) to create a more robust motivational scale that could be validated across more diverse samples [40]. Despite these limitations, the FCQ maintains a strong reputation for reliability because of the many studies that have employed it for various ends [41,42,43,44].

### 2.7. Health and Attitudes Scales

The Health and Taste Attitudes Scales has questions on eating behavior and on food choice [45]. The measure was developed to assess consumers’ orientations towards health and hedonic characteristics of foods on the market. It comprises three health-related factors (general health interest, light product interest, and natural product interest) and three taste-related factors (i.e., craving for sweet foods, using food as a reward, and pleasure) [45]. These questions may be used to study motivations concerning health and taste, with an emphasis on purchasing behaviors. A drawback is that the questions are geared more towards beliefs and opinions than FMVs. For example, one of the questions on light product interest is “I believe that light products keep one’s body in good shape”. Results pertaining to questions on beliefs and opinions will not be as useful as those pertaining to FMVs for promoting dietary change, since beliefs and opinions may not be as easily leveraged as motivation.

The focus on consumer behavior, rather than general food motivations, may be helpful for achieving dietary change, despite the discrepancy between purchasing foods and eating them. If long-term eating behavior change is the goal of the diet intervention, then measuring participants’ attitudes as consumers could help to understand the factors underlying food choice. A more informed understanding of why people purchase certain foods could allow researchers and health professionals to engage in decisional factors (e.g., using natural product interest to drive healthier food choices) or address problematic tendencies (e.g., buying food as a reward).

### 2.8. Measure of Food Choice Values

Lyerly and Reeve developed a measure for food choice values (FCV) from Steptoe et al.’s FCQ [30]. The FCV differs in several respects from the FCQ. The health factor on the FCQ is divided into health/weight concern and a factor called “organic” on the FCV. The organic factor includes questions on natural content and ethical concern (e.g., fair trade, sustainable packaging), which are separate factors on the FCQ. The FCV also differentiates between convenience in preparing and consuming foods and convenience of access (financially and physically), which is labelled “accessibility”. Lastly, another factor was added for food safety. Despite these differences, the FCVs could serve a similar purpose as the FCQ for promoting dietary change, because both ask about central reasons for food choice.

### 2.9. Measurement of Ethical Food Motives

The measure includes three factors: (1) ecological welfare, (2) political values, and (3) religion. It was developed as an addendum to the FCQ [31]. These motives for food choice are usually rated as less influential than other motives, such as health, sensory appeal, and price [31]. Therefore, the tool would be of greatest value to researchers using it in conjunction with the FCQ or who are interested in studying these particular factors as indicators of short- or long-term dietary change.

### 2.10. The Motivations to Eat Measure

Jackson et al. developed the Motivations to Eat measure with a focus on psychological, rather than physiological, motivation to eat or not to eat [32]. The assessment includes four categories of motivators for eating: coping, social, compliance, and pleasure. Importantly, the Motivations to Eat Measure focuses on factors that motivate a person to initiate an eating encounter, not what motivates food choice once the decision to eat has been made. The researchers found that disordered eating behaviors (e.g., bingeing, restrictive eating, and purging) were associated specific motivations: coping motivations predicted bingeing, pleasure motivations positively predicted binge eating and negatively predicted restrictive eating, compliance predicted restrictive eating and purging, and social motivations negatively predicted restrictive eating and purging, but positively predicted bingeing. For health professionals interested in investigating disordered eating patterns and fostering positive psychological motivations for eating as a means of improving adherence, the Motivations to Eat Measure is a useful tool. The measure also may help with the identification of individuals at risk for maladaptive eating patterns which, in turn, could remediate these potential derailments.

### 2.11. The Multiple Food Test

The Multiple Food Test measures applied nutrition knowledge as it relates to food choice [33]. The test is an image selection task, where participants select which foods they would choose and then rate the perceived healthiness of those foods [33]. Therefore, the findings can inform health professionals on individuals’ or groups’ propensity for choosing foods they deem healthy, essentially measuring health as a food motive. The findings also can be used to assess individuals’ knowledge of nutrition, if the individuals’ healthiness scores for the different foods are compared to determined healthiness scores provided by knowledgeable professionals. As the test requires that respondents pick one option out of four different foods, individuals with dietary restrictions will select foods that they are able to eat, even if they consider the other options more nutritious or enjoyable. Therefore, there are two considerations with the Multiple Food Test: (1) it measures individuals’ likelihood of picking a food based on perceived healthiness of the foods, not the actual healthiness and (2) it is not applicable to those with dietary restrictions.

### 2.12. Palatable Eating Motives Scale

The Palatable Eating Motives Scale (PEMS) is useful for analyzing four different motives for eating highly appetizing foods with low nutritional value [34]. The measure was adapted from the Drinking Motives Questionnaire Revised (DMQ-R), which also includes social, conformity, enhancement, and coping motives [34]. Identifying high scores on one or more of the scales can inform individualized advice or group-based interventions. Since cut-off points have yet to be determined, administering the PEMS and relating subset scores to diet adherence and outcomes can further the literature and add to the tool’s usefulness in future studies.

### 2.13. The Three Factor Eating Questionnaire (TEFQ)

The TEFQ measures cognitive restraint, uncontrolled eating, and emotional eating [23]. Cognitive restraint is an individual’s tendency to control what or how much food is consumed. Uncontrolled eating is also described as disinhibition of control, where eating episodes are unrestrained. Emotional eating can be understood as susceptibility to hunger or non-hunger, appetitive cues. The TEFQ was designed to differentiate between those with contrasting behaviors (i.e., dieters and ad libitum eaters) and weight (e.g., BMI and change in weight) [46,47].

With the focus on behavior, the TEFQ does not measure motives or values related to food directly. However, underlying motivations may be tied to the behaviors the TEFQ does measure. For example, cognitive restraint is likely related to motivation for weight control. The behavioral focus may also make results from the TEFQ useful in interventions. When Stunkard and Messick developed the tool, they theorized that those with high cognitive restraint scores may be more responsive to information, whereas those with high uncontrolled eating and emotional eating scores would benefit from behavioral management strategies [23]. Therefore, the TEFQ may be useful for tailoring counselling and interventions.

### 2.14. Comparing Measures

Numerous measures have been developed to assess motivations in eating behavior and food selection, each with their own advantages and disadvantages. Here, we review (1) measures related to eating behavior, (2) tools with specific focuses, and (3) measures capturing broad motivations for food choice.

The Motivations to Eat Measure assesses four core motives (to enhance pleasure, to cope with negative affect, to be social, and to comply with others’ expectations) [32]. It is more concise than the DEBQ. The DEBQ includes measures of emotional eating and eating in response to external sensory cues such as the smell and appearance of food (external eating) as two core motivations to eat [24]. Similarly, the Health and Taste Attitudes Questionnaire quantifies the initiation of eating for the pleasure of taste and choice of foods for health reasons [29]. TEMS, developed by Renner et al., is similar, but more encompassing; it includes other motives for food choice, such as social and physiological motives [25]. The TEFQ focuses on behaviors related to weight control more broadly. These questionnaires are useful in their own right, but generally de-emphasize items related to explicit motivations and values, which may be more pertinent for measuring and leveraging in dietary change.

There are measures that focus narrowly on what category of FMVs, and those that attempt to capture a wide array of motivations. One of those with a narrow focus is the Measurement of Ethical Food Motives, which assess only several less-prevalent factors for food choice related to moral imperatives [31]. The Food and Beverage Uniqueness Scale is another example, with a specific focus on motivations to try novel food items [27]. The Palatable Eating Motives Scale concerns hedonic motives for eating, while the Multiple Food Test analyzes health in addition to tastiness [33,34]. The Health and Taste Attitudes Scales likewise assesses the role of healthiness and palatability [29]. The Food Choice Motives Questionnaire falls in-between measures with narrow emphasis and broad emphasis, with a concentration on sustainability but items related to several other factors as well [48].

The EATMOT Scale, the FCQ, and the FCV are the most encompassing tools [22,26,30]. Although each measure has distinctive benefits and drawbacks, the FCQ developed by Steptoe et al., the modified version with FCVs conceived by Lylerly and Renner, and the EATMOT Scale are perhaps the most applicable measures for food choice motives, encompassing a variety of different food choice motives for everyday life [22,25].

## 3. Previous and Potential Future Uses of Food Motives and Values Assessments

### 3.1. Previous Uses

Much of the work done with food motives focuses on consumers’ purchasing behavior. For example, the Health and Taste Attitudes Scales was developed to assess consumers’ orientations toward the health and hedonic characteristics of foods [45]. Consumer research also has revealed connections between healthy food choices and buyers’ values, expectations, and intentions [49]. Similar work has examined consumers’ relationship with food [50]. In addition to consumers’ internal motives and values, studies have also been conducted to investigate consumers’ patterns of behavior, such as millennials’ motivation and choice to dine-in or take-out [51]. The results inform marketing strategies for restaurants, grocery stores, and health food products [50,51,52]. In essence, the focus on values and motives in individual consumer settings has been useful for informing behavioral change. Although dietary interventions focus on health behaviors, and consumer studies focus on purchasing behavior, the findings from studies on consumer behavior have potential value to researchers studying health behaviors. The lesson may be that paying attention to motivations and values pays off.

Where consumer research has employed food values and motives to successfully achieve behavior change, efforts for dietary change may have fallen short by incorrectly assuming or ascribing certain motives and values, rather than measuring, studying, and making use of individuals’ own motives and values. Measures of food choice have been used in dietary interventions, though the focus is generally on assumed motives (e.g., improving health) or changing motivations, rather than measuring and leveraging previously held motives [53,54]. A central aim of this paper is to initiate change towards becoming more mindful of individuals’ and groups’ motivations for food choice and eating.

In the next sections, we provide examples of the previous uses of several of the questionnaires presented above. We present three main uses of the questionnaires in the past: personalizing the intervention for individuals, adjusting the delivery protocol to align with group values/motives, and identifying factors associated with success (either with adherence or health outcomes).

#### 3.1.1. Individualization

In the first example, researchers interested in personalizing treatments conducted a study asking participants to record what they ate using a food diary for a week. From the food diary, participants gleaned personal motivations for unhealthy snacking [55]. The motivational cues reported for unhealthy snacking were based on the four categories of motivations for eating (coping, social, compliance, and pleasure) described by Jackson et al. in the Motivations to Eat measure [55]. Then, these specified personal motivations for unhealthy eating were paired with implementation intentions for healthier eating, and successful behavior change was achieved [55].

#### 3.1.2. Study Design

Food motives also may be useful for adapting studies within different groups. Ohly et al. measured parents’ food motives prior to implementing a nutrition intervention in the United Kingdom [56]. They found that parents felt health, taste, freshness, and quality were the more important factors in their food choices [56]. They remarked that the specific types of support parents desired could be identified as a result of administering the FCQ [56]. In the US, a team of researchers administered the FCQ to inform the design of a health retail intervention for rural-residing Native Americans [57]. This work, by Wetherhill et al., provides implications for health interventions and public health initiatives [57]. The authors found that many Native-American health interventions emphasized values other than those denoted important by the FCQ, and that health promotion strategies tend to focus on health rather than the relevant dimensions of FCVs in the target population. Their results suggest that healthy food interventions that incorporate these FCVs may be more effective than those that focus on nutrition/health and disregard prior motives and values.

Ohly et al.’s use of food choice to design a nutrition intervention at children’s centers in the United Kingdom and Wetherhill et al.’s use of the FCQ to design an appropriate public health intervention for a population of Native Americans show that the measurement of food values and motives can inform study design for participants in diverse populations [56,57].

#### 3.1.3. Data Analysis

While useful for study design, measures for food choice also can be used to explain results. For example, scores on the FCQ and the DEBQ have been shown to predict weight loss and various other health outcomes in research studies [22,24,58]. Some of the researchers used the FCQ to control for differences between treatment and control groups and to track changes over time, finding that post-intervention scores for health, convenience, sensory appeal and the natural content of food on the FCQ increased relative to baseline, though these changes did not explain differences in BMI [59].

#### 3.1.4. Summary

We identified three ways that these surveys have been employed: (1) using individual motives to provide personalized advice, (2) by identifying predominant group motive(s) to modify an intervention, and (3) by finding motives associated with poor outcomes.

### 3.2. Potential Future Uses: Within Research Settings

There are practical implications for food motives and values in interventions. If the majority of participants rate one or several food motives highly, then educational efforts could be informed accordingly [56,57]. Consider the following hypotheticals. If results from the FCQ (or related measure) at baseline reveal that most of the participants rated convenience as a primary determinant of their food choice, then early intervention sessions could be formatted around healthy convenience foods and easy methods of preparation. If multiple food motives emerge as significant, then intervention sessions could focus on motives sequentially or attempt to integrate multiple motives. Perhaps cost and sensory appeal were the two most important factors; then, guidelines for choosing affordable foods and cooking methods (emphasizing flavors, textures, etc.) could be presented one after the other or concurrently. If participants have a low mean score on familiarity then combining ingredients, including spices, creatively can add novelty while reducing the inflammatory potential of foods [10]. Although the FCQ and other surveys have been used in group and community settings in the past (as with the examples mentioned above), this is a novel suggestion for how these research tools should be employed in dietary interventions.

In addition to using food motives to tailor an intervention to the group, there are other ways of using individuals’ food motives to tailor the intervention to participants experiencing the most difficulty [60]. Continuously monitoring adherence among participants may reveal differences in success with the proscribed diet according to food motives. Perhaps, for example, those who had high emotional eating scores on the TFEQ may be more likely to be non-adherent. Then, intervention sessions could be designed to address emotional eating and provide participants with an opportunity for reflection and self-regulatory strategies. These recommendations are also applicable to health professionals conducting one-on-one counselling.

Food motives can be used to inform individualized advice, in addition to advice for subgroups within a study (namely those struggling with adherence). Participants may be encouraged to contact the dietitian or staff if they have challenges or could benefit from direct attention. In instances where study staff are contacted or contact a participant struggling to adhere to the diet, an understanding of that individual’s food motives can be integral in providing guidance (see Table 2). Health was excluded as a dimension, because dietary interventions already tend to emphasize health [57].

### 3.3. Potential Future Uses: Beyond Interventions

Relating individuals’ diet to broader groups and populations is an area with much potential for discovery. In dietary interventions based in specific communities or for a specific group with a common identity, the participants’ values and motivations may represent the individuals’ social and cultural background. Thus, measuring individuals’ reasons for eating what they eat can provide useful insight as to where motivations and values overlap among participants in real-world, non-interventional settings. Information on commonalities within groups can be used to inform dietary change efforts at the population level.

Syndemic theory is useful for bridging individual and population-level perspectives. The idea of syndemics centers on the clustering of diseases within groups of people and pays particular attention to the influence of joint biological, social, and psychological determinants [7]. Macro-level health determinants are an important component of diet-related diseases [7]. These psychological tools that measure FMVs can advance our understanding of the personal and shared values related to health outcomes. Furthermore, results from these measures can afford important perspective on distinct sociocultural challenges related to nutrition.

Understanding why people eat what they eat can provide a deep understanding about the lives they live. Where one person may be struggling with ideas for recipes, another may be struggling with anxiety and overeating, and a third individual may be struggling with food insecurity. Food motives and values can provide researchers and health professionals with a deeper perspective on who the participants/clients are: what they care the most about, specific challenges in their lives, and where the opportunities for intervention lie. If health professionals assume that health is the individual’s primary concern and center the intervention or messaging around health-related facts, then a valuable opportunity may be missed to provide information or advice that aligns with the individuals’ priorities. To say that individuals value time, or money, or the environment is not to say that they do not also value health when making decisions related to food. Nevertheless, health professionals must ask themselves if the information being provided is the information that will have the most significant impact on individuals’ health status. Asking questions is essential to avoiding unwarranted generalizations or other assumptions. Table 2 outlines examples of how FMVs (from Steptoe et al.’s FCQ) can be used in individual counselling [22]. Uncovering where and why dietary change is unsuccessful will inevitably be tied to intrapersonal and interpersonal constraints and challenges. Understanding trends in motivations, values, and challenges derived from community-based surveying can provide another layer of meaningful information. For example, if a group of college students is surveyed using the DEBQ, and the findings denote high levels of emotional eating, that information can be used to develop effective health messaging and behavioral interventions that specifically relate to the psychopathology underlying dietary issues. In this case, data can draw attention to clustering of unhealthy behaviors and disease. Findings can also deepen our understanding of social factors in diet-related disease. Social norms contributing to unhealthy behaviors, such as uncontrolled eating, could be studied using measures such as the TEFQ. These tools are useful for understanding the “whys” behind socio-cultural factors affecting food and eating behavior. Ultimately, findings can inform actions and recommendations on a larger scale.

## 4. Conclusions

The descriptions, strengths, limitations, and implications for research measures related to eating behavior, food motives and values provide researchers and health professionals with a basis for selecting apt tools to meet specified goals. We provided a description of novel directions for the future of dietary counselling, research, and recommendations, which are summarized in Table 3.

There are limitations to our suggestions. First, this perspective is based on a careful search of the literature that is not, technically, systematic; neither is the article a scoping or narrative review. Our goal was to present a perspective on the importance of FMVs, which necessitated a fair selection of measurers and not necessarily a review of the literature. Future studies may aim to expand on our perspective by conducting a systematic review of the research. Similar measures, such as the Emotional Eating Scale and the Eating Behavior Inventory, could also be included in a future review.

The second limitation concerns our recommendations about individualized treatment in dietary interventions. The concern is that by providing individualized treatment, a dietary intervention, which is intended to be controlled, will lose its experimental validity. Although the opportunity for personalized counseling should be the same for the treatment and control groups, one consideration is that it is possible that more individuals from one group could utilize these resources, thereby undermining the attempt to balance using the randomized study design [61]. However, ideas contained in this perspective could be incorporated into the design of adaptive trials that take into account individual patient preferences, which is further explored in another article [19]. While the RCT paradigm has been considered the strongest study design in biomedicine, there are definite benefits for “operationalized and individually tailored strategies for prevention and treatment of chronic, relapsing disorders” [17,62,63]. We believe that food values and motives can be effectively used in adaptive dietary interventions to improve adherence and participants’ outcomes, even if such adaptations seem to compromise the rigor of the study design.

Despite the potential drawbacks, FMVs have the potential to reshape efforts for dietary change. In addition to personalization in research and clinical settings, measures of food choice may reveal that particular food motives or groupings of food motives are associated with successful dietary change, which could inform health professionals’ and researchers’ future efforts [18]. Next, results from these measures can be used to analyze correlates of adherence and positive outcomes. Collectively, findings can contribute meaningfully to the literature on FMVs. Finally, best-practice public health nutrition education, intervention, and recommendations can benefit from work on FMVs with individuals, groups, communities, and populations.

## Figures and Tables

**Table 1 nutrients-13-01452-t001:** Measures of food motives and values.

Measure	Authors	Description	Strengths	Shortcomings	Applications
1. Dutch Eating Behavior Questionnaire(DEBQ)	van Strien et al., 1986 [24]	Includes three scales on eating proclivities and behaviors. Developed for use in overweight and obese populations.	Useful for identifying behaviors associated with weight-related health outcomes, including obesity and anorexia nervosa.	Not applicable to individuals without disordered eating patterns.	Identification of maladaptive eating patterns for individualized treatment. Use to inform study design. Use in post hoc analyses to determine relationships between eating behaviors and adherence/outcomes.
2. The Eating Motivation Survey (TEMS)	Renner et al., 2012 [25]	Measures motives for food choice in the general population. Focuses specifically on the pathology of adaptive eating behaviors (rather than maladaptive/disordered eating patterns).	Comprehensive; includes motives from various other measures as well as from nutritionist interviews, discussion groups by psychologists, and the authors’ input.	Developed in Germany, may have limited usefulness in diverse populations (racial-ethnic demographic characteristics of the study population not reported).	Can be used for individuals or groups, but most applicable in Western populations, which may be overweight, but do not exhibit disordered eating behaviors. Use results to tailor individual dietary counselling, dietary interventions, and public health messages and campaigns to specific food choice motives/values.
3. The Eating Motivations Scale (EATMOT)	Raquel et al., 2020 [26]	An expanded, internationally developed and tested measure for food motives.	Widely applicable. Can be used to identify food choice motives in different geographical areas. Both concise and encompassing.	May be less useful than other measures in subpopulations where specific motives are believed to play a significant role.	Provide basis for individualized advice; widely applicable. Inform study design to focus on particular motives. Use in post hoc analysis to elucidate correlates of dietary adherence/non-adherence and outcomes. Use in communities and populations to tailor health messaging to food motives and values.
4. Food and Beverage Need for Uniqueness Scale	Cardello et al., 2019 [27]	Trait scale that measures individuals’ propensity for unfamiliar foods and beverages.	Indirectly measures novelty as a motive for food choice. Developed for use internationally.	Does not encompass motives for food choice beyond uniqueness.	Findings may suggest prioritization of novel foods and recipes for individual dietary counseling, in interventions, and for public health nutrition messaging.
5. Food Choice Motives Questionnaire	Sautron et al., 2015 [28]	Measures food choice motives for purchasing, particularly those relating to product sustainability. Developed primarily to assess consumer motives and promote sustainable diets.	Includes nine broad dimensions. Several of the dimensions are not included in other questionnaires, such as local and traditional production and innovation.	May provide unhelpful insight, since the focus is on consumer behavior related to sustainability rather than eating behaviors/food motives generally.	Identifying and characterizing a populations’ concern about sustainability. Measure reasons for food choices during purchasing. Useful for direction informational campaigns specifically aiming to increase the purchase and consumption of environmentally friendly food products.
6. Food Choice Questionnaire (FCQ)	Steptoe, Pollard, and Wardle, 1995 [22]	Among the first questionnaires on food motives to be introduced. Measures 9 factors related to motives for food choice.	Useful for identifying categorical motives for food choice. Widely utilized, validated, reliable.	Lack of cross-cultural validation. Subject to social desirability response bias. Lack of comprehensiveness. Inappropriate abstraction.	Provide basis for individualized advice. Inform study design to focus on particular motives. Use in post hoc analysis to elucidate correlates of dietary adherence/non-adherence and outcomes. Use in communities and populations to tailor health messaging to food motives and values.
7. Health and Taste Attitudes Scales	Roininen, Lahteemnaki, and Tuorila, 1999 [29]	Encompasses items on eating behavior and food choice to assess consumers’ orientations towards health and hedonic characteristics of foods on the market.	Evaluates various facets of both health and taste. Provides more detail on these factors than the FCQ. Items also assess behaviors that seem to be common barriers to dietary change (e.g., cravings, using food as a reward).	Not concerned with motivations and values so much as beliefs and opinions, which may be more indirect measures of individuals’ food choices.	Provides information on individuals’ orientation towards health or hedonic characteristics of food. Studying how these motives change over time and relate to success in dietary change can provide greater insight in food psychology.
8. Measure of Food Choice Values (FCV)	Lyerly and Reeve, 2015 [30]	Developed from the FCQ. Principal changes include reorganization of “health” category and the differentiation of convenience of access and preparation.	More recent than the FCQ. Incorporates additional food choice values (FCVs) not present in the FCQ. Initial studies indicate results are not affected by social desirability. Predictive of dietary intake.	Fewer validation and reliability studies than the FCQ. Not tested in as many diverse populations at the FCQ.	Provide basis for individualized advice. Inform study design to focus on particular motives. Use in post hoc analysis to elucidate correlates of dietary adherence/non-adherence and outcomes. Use in communities and populations to tailor health messaging to food motives and values.
9. Measurement of Ethical Food Motives	Lindeman and Väänänen, 2000 [31]	Developed from previous studies on vegetarianism and the influence of ideology and religion on food choice. Focuses specifically on ethical motivations.	For some individuals and groups, ethical motives may override other food choice motives, so measuring and using ethical motivations could provide stronger impetus for dietary change.	Ecological welfare, political values, and religion may be rated as less important than many other motives, such as health, sensory appeal, or price. The measure may not be useful for the general population in capitalistic, secular countries.	Use as an addendum to the FCQ/FCV in subgroups with overt ethical motives for food choice. Results can be used to set priorities in dietary interventions and tailor information delivery.
10. The Motivations to Eat Measure	Jackson et al., 2003 [32]	Measures psychological motivations to eat or to abstain, including coping, social, compliance, and pleasure motivations.	Assesses motivating factors for initiating an eating encounter (when to eat) rather than which foods are selected (what to eat). Associated with disordered eating behaviors.	Does not provide insight on the selection of food types or preparation methods.	Identification of maladaptive eating patterns. Use to recognize barriers to successful dietary change and/or weight loss in individuals, participants, or subgroups. Use in post hoc analyses to determine relationships between eating behaviors and adherence/outcomes.
11. Multiple Food Test	Schreiber et al., 2020 [33]	Image selection task based on food choice among four options and subsequent ranking of their healthiness.	Useful for determining the role of applied nutrition knowledge on food choices, which can provide insight on the importance of health in individuals’ food choice.	Focuses narrowly on the moderating role of nutrition knowledge in food choice. Results are not applicable to individuals with food intolerances, allergies, or dietary restrictions.	May provide insight to dieticians and researchers about individuals’ applied nutrition knowledge and the role of that knowledge in food choice. These insights may be useful for providing nutritional education, but the participants’ other food motives and values should be considered (using other tools) before doing so.
12. Palatable Eating Motives Scale	Burgess et al., 2014 [34]	Adapted from the Drinking Motives Questionnaire Revised. Quantifies four different motives for hedonic eating behaviors.	Provides insight on unique motives for eating palatable foods, which can undermine dietary progress.	Only includes motivations for eating palatable foods (sugary foods and drinks, fast foods, and salty foods). Useful for individuals/groups who explicitly struggle with food cravings. Clinically useful cut-off scores not yet determined.	Identification of maladaptive eating patterns. Use to recognize barriers to successful dietary change and/or weight loss in individuals, participants, or subgroups. Use in post hoc analyses to determine relationships between eating behaviors and adherence/outcomes. Further literature on food motives and eating behaviors.
13. The Three Factor Eating Questionnaire(TFEQ)	Stunkard and Messick, 1985 [23]	The TFEQ assess three psychological factors related to eating behavior: cognitive restraint, uncontrolled eating, and emotional eating. The measure was designed to differentiate dieters and ad libitum eaters.	Scores on the first factor can provide insight on motivation to eat, while the second and third factors give information on behavioral challenges related to adherence and success.	The measure does not assess motives or values pertaining to food choice directly but does provide insight on how individuals eat.	Provide basis for individualized advice. Those with high factor I scores are likely to be more responsive to information, while those with higher factor II and III scores may gain more from behavioral intervention. Use in post hoc analyses to determine relationships between eating behaviors and adherence/outcomes.

Summary of the strengths, weaknesses, and applications of important measures related to food motives and eating behaviors.

**Table 2 nutrients-13-01452-t002:** Potential use of food motives and values for individual support.

Motive Identified from Questionnaire	Examples of Potential Challenges	Questions and Solutions
Cost	The individual is having trouble paying for fruits and vegetables.	Suggest trying frozen fruits and vegetables. Ask if he/she is aware of or has considered applying for the Supplemental Nutrition Assistance Program. Ask if his/her regular grocery store has a discounted produce rack and/or coupons.
Mood	The individual usually struggles to adhere to the diet in the evenings after coming home from work feeling emotionally drained.	Ask about emotional triggers that precede non-adherent episodes. Brainstorm ideas to mitigate stressful situations. Suggest techniques to help participant be mindful of their mood prior, during, and after eating. Suggest alternatives to eating (i.e., journaling, taking a walk, listening to music, etc.) Ask him/her to record how they feel after every meal.
Convenience	The individual is working two jobs, one of which requires an extensive amount of driving in the evenings, late into the night.	Ask what he/she normally eats on the road. Brainstorm ideas for preparing foods beforehand. Discuss preferable options when on the go (e.g., grocery store salad bars, any treatment-compliant options at fast food restaurants).
Sensory Appeal	The individual feels that foods prepared in the intervention sessions (cooking classes) are not appealing to his/her palate, saying salty, sweet, and crunchy is the combination he/she usually seeks.	Ask what foods and snacks the participant ate prior to the study. Ask about preferred foods and qualities of those foods that were appealing. Propose ideas that agree with his/her inclinations. Gauge his/her liking or disliking of each in offering further suggestions.
Natural Content	The individual who cooks for the family is having difficulties cooking foods that satisfy everyone’s preferences.	Ask him/her about the challenges they are experiencing with competing motives (e.g., he/she is motivated by natural content and family members are not). Suggest he/she sit down with a recipe book and family members and find meals they agree upon.
Weight Control	An individual who previously followed a low-carbohydrate diet is concerned that eating more complex carbohydrates (grains, legumes, starchy vegetables, etc.) is causing weight gain.	Listen to his/her concerns intently. After validating his/her feelings, inquire about motives for joining a dietary intervention (perhaps the previous diet had its own shortfalls that motivated the participant to try something new). Tread carefully when deciding whether to describe findings from studies that suggest whole grains, legumes, starchy vegetables, and the like are not associated with weight gain. Opt to offer anecdotes that will help the participant feel understood and encouraged rather than undermined and discouraged.
Familiarity	The individual is unsure what to eat in place of regular snacks and meals and says he/she feels lost trying to follow the new diet.	Ask what he/she regularly ate before joining the study. Ask what he/she ate growing up. Ask what new foods that he/she has tried or would be willing to try. Use these data to formulate suggestions for foods that fall within the diet guidelines.
Ethical Concern	The individual is concerned that the diet will not be in agreement with moral/religious convictions.	Suggest alterations that are concordant with ethics and study guidelines (e.g., participant is environmentally conscious; suggest foods that fall within diet guidelines with a smaller carbon footprint).

Eight factors motivating food choice from Steptoe et al.’s Food Choice Questionnaire, along with their respective applications in dietary interventions or dietary counselling.

**Table 3 nutrients-13-01452-t003:** Potential uses of food motives and values measures summarized.

Realm	Specific Uses
Individual Support	Provide informed and personalized advice for challenges to address non-adherence.
Study Design	Informing interventions to focus on particular food motives. Providing data to tailor future interventions.
Data Analysis	Food choice could be a potentially important variable to examine in treatment and control groups as a confounding or moderating variable. Food choice may be a predictor variable of adherence/non-adherence or health outcomes.
Public Health Education, Interventions, and Recommendations	Motives for food choice may be a source of population-level nutritional disparities. Tailoring public health messages and educational campaigns to specific food choice values. Meaningfully informing community health initiatives and marketing strategies to particular groups. Findings can be used to study clustering of unhealthy behaviors and factors affecting diet-related disease, disease–disease interactions, and social condition–disease interactions.
Furthering Literature on the FMVs	Examining the cultural relativity of the FMVs in different samples. Potential to validate or modify the FMVs in different samples. Investigating change in food choice motives over the course of intervention-type studies. Directing future research questions.

Findings from food motives and values measures can be applied across public health dimensions, from fields that concern individuals to those that consider populations.

## Data Availability

All results reported here are freely available in published research.

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
