# Peer review of "Measuring and Leveraging Motives and Values in Dietary Interventions"

_nutrients, 2021, doi:10.3390/nu13051452_

Round 1
Reviewer 1 Report
General comments
Measuring and leveraging motives and values in dietary Interventions is a paper that wants to discuss ways to access reasons and uses related to measuring and leveraging food motives and values. However, to develop that, the authors need to describe the comprehensive approach underlined in the study design and data analysis. Bellow, are presented some comments that could be considered in the revision of the material:
Abstract – Must include the research question, study goal and details about the review type method used as well as main findings.
Introduction – The background should mention the conceptual and theoretical frameworks underlying the measurement of motives and values in dietary interventions. Important information that gives support to the method used.
Methods – It is not clear the type of review performed. The research question must be presented. Authors should explain the motives to use "Steptoe, Pollard, and Wardle's Food Choice Questionnaire (FCQ)" material as "seeds" for the search strategy (that need to be well explained), data extraction and data mapping procedures too. The presentation of the framework by a flow chart could be helpful.
Suggested reference:
RETHLEFSEN, Melissa L. et al. PRISMA-S: an extension to the PRISMA Statement for Reporting Literature Searches in Systematic Reviews. Systematic reviews, v. 10, n. 1, p. 1-19, 2021.
TRICCO, Andrea C. et al. PRISMA extension for scoping reviews (PRISMA-ScR): checklist and explanation. Annals of internal medicine, v. 169, n. 7, p. 467-473, 2018.
Author Response
Reviewer #1
Abstract – Must include the research question, study goal and details about the review type method used as well as main findings.
As mentioned previously, it is important to note that this is a perspective not a systematic review. Despite this, we have detailed the method used for reviewing articles as part of this process. Specifically, we have added the following to the Abstract:
“So, this research question naturally arises: How can food motives and values be measured and leveraged to improve diet outcomes from the individual to populations?”
We also modified the text on the lines 22-28 to make clear how we organized this perspective:
“A central aim of our study is to provide specific examples and recommendations on how to measure and leverage FMVs. To do so, we reviewed 12 measures included in the literature citing the Food Choice Questionnaire by Steptoe, Pollard, and Wardle, which was identified as the earliest, highly cited article appearing under the search terms "food motives,” "food values," "eating behavior," AND "measure." Specific details on how articles were selected from the citing literature are descried in the Methods section. We also expound on our reasoning for including the Three-Factor Eating Questionnaire, which made for 13 measures in total.”
Introduction – The background should mention the conceptual and theoretical frameworks underlying the measurement of motives and values in dietary interventions. Important information that gives support to the method used.
To the Introduction we have included:
“Why measure food motives to address long term dietary change and low adherence? And how? We can measure FMV's by examining explicit reasons for food choice or implicit influences on eating behavior (using the tools in the following section). Studying these factors is likely to be useful in understanding why some individuals fail to improve their diet, while others succeed (in research and real-world settings).”
We believe that this is now sufficient. However, if there is any more specific conceptual or theoretical clarification we might be able to provide, we would be happy to do so.
Methods – It is not clear the type of review performed. The research question must be presented. Authors should explain the motives to use "Steptoe, Pollard, and Wardle's Food Choice Questionnaire (FCQ)" material as "seeds" for the search strategy (that need to be well explained), data extraction and data mapping procedures too. The presentation of the framework by a flow chart could be helpful.
We further clarified our methods by saying:
“Because this article was intended to be a perspective on the appropriate use of measures of FMVs, and not to provide a systematic review of the literature meant to identify and catalog all articles addressing any aspect of food motivation or values we approached this as follows. We first searched Clarivate Web of Science® for the terms "food motives” AND "food values" AND "eating behavior" AND "measure."
“From this, we found the earliest measure of FMVs was Steptoe, Pollard, and Wardle's Food Choice Questionnaire (FCQ, published in 1995). This highly cited article (914 of 979 total citations as of 9 April 2021 according to Web of Science), enabled us to use the FCQ article as a “seed” article [22]. We then searched the titles of citing literature (i.e., articles referencing the FCQ) for the following terms: "measure," "tool," "questionnaire," "survey," OR "scale,” which provided 818 articles.”
“From 80 articles that potentially furthered the study aim to describe measures of FMVs and the applications, we identified 12 measures for inclusion. The Three Factor Eating Questionnaire (TFEQ) does not cite the FCQ, but because the measure occupies an important place in the literature (it has been cited 3019 times as of 9 April 2021, it also was included.”
We decided against the inclusion of a flowchart, which are commonly presented in meta-analyses, systematic reviews, and similar studies and, therefore, would be inappropriate here.

Reviewer 2 Report
This is the second time I have been asked to review your paper. I am happy to see that the Authors have greatly ameliorated their MS and took all previous comments into absolute consideration.
I, therefore, consider it as a revision and have no further comments.
Author Response
Thank you for your comments.
Round 2
Reviewer 1 Report
I am happy to see that the Authors have agreed with the comments. I have no further consideration.
This manuscript is a resubmission of an earlier submission. The following is a list of the peer review reports and author responses from that submission.
Round 1
Reviewer 1 Report
This is an exciting manuscript which addresses a review of strengths and weaknesses among the different measures of motives and values that affect food choice, apart from the relevance of this topic, some methodological issues need to be considered. As presented below:
Authors indicated that they reviewed individual strengths and weaknesses among measures of motives and values that affect food choice but do not mention how they conducted the review. Detailed methodological aspects included in the review process must be presented.
The dietary recommendation must be population-based. Social and cultural structures shape individual and group motives and values, relevant statement when we are measuring and leveraging motives and values in dietary Interventions. Indeed, these aspects need more profound discussion to align individual and group perspectives and local characteristics. Topics that also should be presented in the debate. Furthermore, the view of syndemics as a new path for global health research also need to be considered in your paper.
Suggested reference:
Mendenhall, E. Syndemics: a new path for global health research. The Lancet 2017; 389(10072): 889-891.
Systematic Review Standards & Organizations available at https://www.nihlibrary.nih.gov/services/systematic-review-service/systematic-review-standards-organizations accessed on December 28th, 2020.
Reviewer 2 Report
Dear authors,
The paper is a narrative review of various tools used in measuring nutritional behavioral related aspects. these are covered mostly by Behavioral & Cognitive models.
Although personalized nutrition is recommended, how do authors recommend interventions to be performed based on suggested subjective recommendations?
Interventions are group-based and more subjective measures - not completely open ended should be used in order to be usable.
Reviewer 3 Report
Some tools used to measure consumers’ food choice motivations, values, emotions etc. are presented in this paper. They are briefly described.
Also, previous and potential uses of these tools are suggested, but without theoretical support.
In sum, no tools are used to carry on a rigorous bibliometric analysis (i.e. Scimat software).
This is just a very interesting, but simple, descriptive study where some tools are described.
Managerial implications have not been provided.